# Water Saturation Prediction in the Middle Bakken Formation Using Machine Learning

Ilyas Mellal [1], Abdeljalil Latrach [1], Vamegh Rasouli [1], Omar Bakelli [2], Abdesselem Dehdouh [1] and Habib Ouadi [2,*]

1 Department of Energy & Petroleum Engineering, University of Wyoming, Laramie, WY 82071, USA; imellal@uwyo.edu (I.M.); alatrach@uwyo.edu (A.L.); vrasouli@uwyo.edu (V.R.); adehdouh@uwyo.edu (A.D.)
2 Department Petroleum Engineering, University of North Dakota, Grand Forks, ND 58202, USA; omar.bakelli@und.edu
* Correspondence: habib.ouadi@und.edu; Tel.: +1-701-215-8911

**Abstract:** Tight reservoirs around the world contain a significant volume of hydrocarbons; however, the heterogeneity of these reservoirs limits the recovery of the original oil in place to less than 20%. Accurate characterization is therefore needed to understand variations in reservoir properties and their effects on production. Water saturation (Sw) has always been challenging to estimate in ultra-tight reservoirs such as the Bakken Formation due to the inaccuracy of resistivity-based methods. While machine learning (ML) has proven to be a powerful tool for predicting rock properties in many tight formations, few studies have been conducted in reservoirs of similar complexity to the Bakken Formation, which is an ultra-tight, multimineral, low-resistivity reservoir. This study presents a workflow for Sw prediction using well logs, core data, and ML algorithms. Logs and core data were gathered from 29 wells drilled in the Bakken Formation. Due to the inaccuracy and lack of robustness of the tried and tested regression models (e.g., linear regression, random forest regression) in predicting Sw as a continuous variable, the problem was reformulated as a classification task. Instead of exact values, the Sw predictions were made in intervals of 10% increments representing 10 classes from 0% to 100%. Gradient boosting and random forest classifiers scored the best classification accuracy, and these two models were used to construct a voting classifier that achieved the best accuracy of 85.53%. The ML model achieved much better accuracy than conventional resistivity-based methods. By conducting this study, we aim to develop a new workflow to improve the prediction of Sw in reservoirs where conventional methods have poor performance.

**Keywords:** petrophysical analysis; Bakken Formation; reserve estimation; tight reservoirs

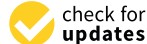



## 1. Introduction

Tight reservoirs hold significant oil and gas reserves; however, their complex rock and fluid properties present production challenges [1]. Despite the application of multistage hydraulic fracturing, less than 20% of the original oil in place can be technically and economically producible [2]. Reservoir simulation results tend to overestimate production from unconventional reservoirs due to uncertainties in characterizing rock and fluid properties variation. Therefore, accurate characterization of these reservoirs is crucial for production forecasting, gas storage, and enhanced oil recovery [3]. One of the significant challenges in estimating Sw in tight reservoirs is the reliance on resistivity-based methods that depend on formation-dependent variables known as Archie parameters [4]. These parameters can be difficult to estimate due to the high variation in the cementing minerals and volumes and to the presence of conductive minerals and high-salinity formation water [5,6]. To address this challenge, ML models have been widely applied in various conventional and complex sandstone and carbonate reservoirs to predict petrophysical properties when conventional methods fail to accurately estimate these properties, as studied by [7,8]. Some

ML algorithms have shown promise in accurately predicting Sw from well logs in tight reservoirs. However, the accuracy of the results is highly dependent on the quality of the well logs used and the complexity of the studied reservoir [9].

Due to the potential to improve the estimation of Sw in complex reservoirs using data-driven models, the application of ML has been the subject of several extensive studies over the past two decades. One of the very first studies on fluid saturation prediction using ML algorithms was conducted by Amiri et al. [10], who compared the Sw results of conventional models, such as the Indonesian and improved Indonesian models, with the Sw predicted from the imperialist competitive algorithm (ICA) artificial neural network (ANN) and backpropagation (BP) ANN models in the tight gas sand of the Mesaverde group located in various basins such as the upper Great River Basin, Piceance Basin, Uinta Basin, and Washakie Basin. Both ANN models scored a correlation coefficient (R2) of more than 0.92 in their study, which was much higher than the R2 of the conventional methods. Boualam [9], on the other hand, conducted a detailed petrophysical analysis of the Three Forks Formation, Williston Basin, which is a thin-bedded carbonate formation, and found that the estimation of Sw using resistivity-based methods was challenging. The author applied two ML modes (support vector regression (SVR) and ANN). Both models achieved correlation coefficients of approximately 0.78, demonstrating higher accuracy compared to the Sw estimates obtained from resistivity-based methods. Miah et al. [11] addressed the problem of predicting fluid saturations in low-resistivity, shaly sand formations using two ANN models, achieving a correlation coefficient of 0.9968. Hodavimoghaddam et al. [12] tested a total of four ML algorithms, XGBoost, LightGBM, CatBoost, and AdaBoost, without relying on the resistivity log. The study was conducted in a tight sandstone reservoir in Russia using triple combo logs from 11 wells. Most of the models provided a very good prediction of Sw.

Despite the promising results found in these studies, most were conducted in either conventional reservoirs with relatively simple lithologies or tight reservoirs with relatively simple mineralogy, pore size distribution, and constant volume of cementing minerals. Additionally, these studies relied on data from a limited number of wells to develop their predictive models, sometimes as low as data from two wells [13], which is not a good starting point for developing a data-driven model capable of generalization and performing predictions on new unseen data from new wells that can potentially have different petrophysical properties.

This study aimed to accurately predict Sw in the MBM using seven classification ML algorithms. It was motivated by poor correlation coefficient results from regression ML algorithms and resistivity-based methods tested on the same sample set. We collected 378 diverse samples from 29 wells in the Bakken Formation across six counties in North Dakota, USA. The dataset included gamma ray, deep resistivity, neutron porosity, bulk density, and Dean–Stark Sw measurements. The aim was to develop a robust ML model capable of capturing maximum variance and mapping the input variables to Sw accurately. This study discusses the results of the implemented ML models and their potential for predicting other petrophysical properties in highly heterogeneous reservoirs and outlines future work to enhance accuracy and generate a Sw map of the MBM. These findings benefit petrophysicists and petroleum engineers, improving the prediction of petrophysical properties and enabling accurate reservoir characterization and simulation in unconventional reservoirs.

## 2. Geological Settings

The Middle Bakken Member (MBM), as a member of the Bakken Petroleum System, is the main producing unit in the Williston Basin and the second oil-producing formation in the United States [14,15]. The formation is located between the Upper and Lower Bakken shale members, with a thickness ranging from 20 to 50 feet in most areas [16]. The geology of the MBM is complex and heterogeneous, with variability in mineralogy, pore types and distribution, and permeability along the formation [17]. The thin layers, bioturbations,

and the variation in lithofacies (which can be up to eight lithofacies) of the MBM result in low reservoir quality despite the high volume of hydrocarbons. This presents a significant challenge for extracting oil and gas [18]. Besides the tightness and the geologic complexity of the formation, the MBM has a low resistivity reading despite the presence of a high volume of oil. Unfortunately, well logs cannot depict the variation observed in Figure 1 due to the low resolution of the logging tools and the high heterogeneity of the petrophysical and geological properties, which vary from one lithofacies to another [18,19]. Mineralogy also plays a substantial role in the characterization challenges [20–22] as the MBM has up to eight minerals, which consist of quartz, calcite, dolomite, illite, kaolinite, K-Feldspar, Muscovite, and plagioclase, where illite, kaolinite, dolomite, and calcite play the role of the cementing materials. This makes the characterization of the mineralogy along with the Sw challenging and the use of conventional methods inaccurate due to the linear models used for mineralogy estimation and the dependence of the Archie equation on the lithology variation [21]. Figure 1 shows the geological feature variation of the five lithofacies of the studied well in the MBM [23].

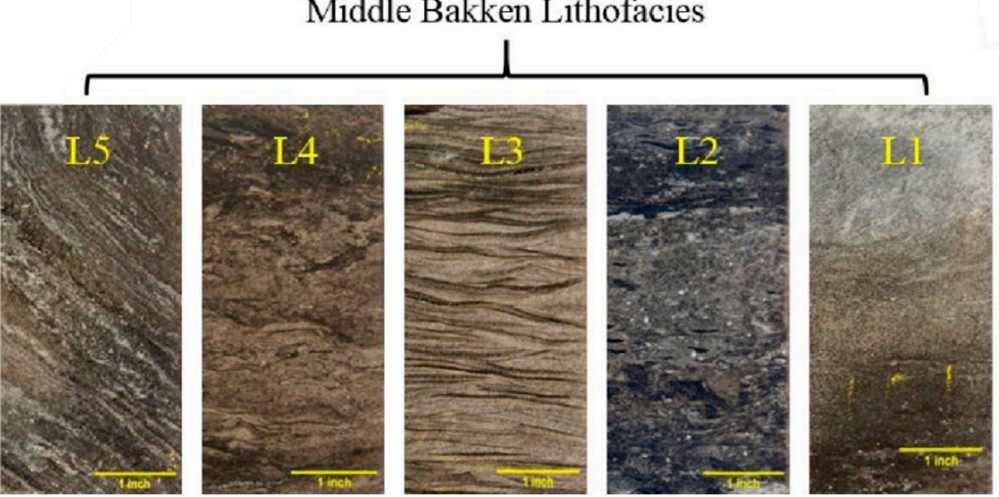

**Figure 1.** Pictures of cores taken from different Middle Bakken facies. L1, L2, L3, L4, and L5 represent the MBM lithofacies [23].

### 3. Materials and Methods

In this study, a total of 378 data points were collected from 29 wells drilled in the MBM. The datasets used to predict Sw consist of conventional logs (gamma ray, depth resistivity, bulk density, and neutron porosity), and Sw was calculated using the modified Simandoux equation, while the Dean–Stark Sw was used as the dependent variable in the dataset. These data were split into 80% training and 20% testing sets and used to train and evaluate the accuracy of different ML models. The best-performing model was then compared to conventional resistivity-based Sw estimation methods.

### 3.1. Petrophysical Data Processing

To accurately predict Sw using ML models, we selected well log values at the same depth as core data, as core plugs provide point values while well logs are continuous. Before conducting this study, petrophysical pre-processing of the dataset were performed. This involves three key steps, environmental correction, depth shifting, and log normalization. The objective of this phase was to generate a comprehensive and coherent set of continuous log and core data curves.

Firstly, environmental corrections were performed to correct resistivity, bulk density, and neutron logs for factors such as borehole properties, mud characteristics, temperature, pressure, and salinity. Secondly, block-shifting was applied to preserve core data integrity. Finally, log normalization ensured consistent analysis results across the selected wells,

avoiding significant deviations in calculations such as clay volume from gamma ray logs. This comprehensive processing approach lays the foundation for an accurate prediction of Sw in the MBM.

*3.2. Machine Learning Model Description*

Seven classification ML models were selected, tested, and compared. The initial aim of this study was to formulate a regression problem and attempt to predict the exact values of Sw. However, after trying several models, refining them, and optimizing the parameters, the models gave poor results, with the best performer, the random forest regressor, only giving an R2 score of 0.53. Instead of a regression problem, the continuous saturation data were converted into 10 classes of 10% saturation intervals, and classification models were used instead. This approach takes into account the uncertainty of the Dean–Stark Sw measurement, accounting for potential over/underestimation caused by salt precipitation or incomplete solvent drying [24]. It is, therefore, a tradeoff between accuracy and precision; using classes with 10% saturation intervals indeed yielded good prediction accuracy, but we traded off the precision of these predictions.

3.2.1. Logistic Regression

Logistic regression is a supervised classification model that uses a linear combination of the input variables followed by a logistic function (i.e., sigmoid). Logistic regression is a binary classification model, with the dependent variable being binary (i.e., either 0 or 1). Nevertheless, it can be extended to K cases to support multiclass classification [25].

3.2.2. Support Vector Classifier

The support vector classifier (SVC), or support vector machine (SVM) classifier, is another supervised learning classification algorithm. SVC works by finding an optimal hyperplane in a hyperdimensional space that separates the different classes of our classification problem. This hyperplane is found by maximizing the distance between this hyperplane and its closest point from each class [26].

3.2.3. Random Forest Classifier

The random forest classifier is an ensemble model that uses predictions from multiple trained decision trees to develop more robust and accurate predictions. The decision trees that make up the random forest classifier are each trained on a random subset of the data, then used to predict the classes of new data. The aggregate of the trees' predictions is the outcome of the random forest [27,28].

3.2.4. AdaBoost

AdaBoost, or adaptive boosting, is another type of ensemble model that iteratively combines weak learners to achieve a stronger learner. After each iteration, wrongly predicted samples are weighted stronger than the rest (i.e., it modifies the sample distribution) to emphasize them more. Although AdaBoost also uses decision trees (like random forest), the different decision trees have different weights on the final prediction, with the weights assigned based on their performance, and they are not treated equally like in random forest [29,30].

3.2.5. Gradient Boosting Classifier

Gradient boosting is yet another type of ensemble method. It is different from AdaBoost in the way it adds and trains the weak learners; it does not change the distribution of the sample and instead trains the weak learner on the residual error of the previous learner. Gradient boosting also employs a gradient-based optimization scheme for a differentiable loss function [31].

### 3.2.6. Artificial Neural Networks

Artificial neural networks are types of models inspired by the structure of biological neurons. They consist of several layers: (1) input, (2) hidden, and (3) output layers. Each layer contains a specific number of nodes or neurons. Data are fed into the model through the input layer and move forward in the network, where each layer performs matrix multiplication followed by a nonlinear activation function and passes the output to the next layer until the output layer is reached, as shown by [32]. The matrices used for computation contain the parameters of the network or weights. The whole process of moving forward in the network from input to output is termed forward propagation. During the learning phase, the weights are initialized randomly, and the network output is calculated. A loss function is used to measure the prediction error; then, the backpropagation algorithm is used to perform auto differentiation on the loss function with respect to the model's weights. Gradient descent is used to optimize these weights to reduce the model's error. This process goes on until a minimal loss is achieved [33,34].

For this study, we used a neural network with 8 inputs (i.e., well logs and calculated inputs), a hidden layer with 32 neurons, and an output layer with 1 output and Softmax activation function (i.e., multiclass classification).

### 3.2.7. Voting Classifier

The voting classifier is an ensemble ML model that takes the predictions made by several separately trained estimators (e.g., SVC, random forests) and aggregates these predictions using two types of voting systems:

- Hard voting: the final prediction is based on the majority vote of the estimators.
- Soft voting: the final prediction is the average of the class probabilities predicted by each model.

The voting classifier does not perform classification but draws on the combined predictions of several models to obtain a more accurate estimate [35].

### 3.3. Data Scaling

Data scaling is an important step prior to feeding data into ML models [36]. Scaling the input variables affects the model performance and interpretability in several ways:

- Improved model performance by reducing the effect of variables' differences in scale.
- Faster model convergence, especially for neural networks with gradient descent optimization.
- Better interpretability by making it easier to compare the different coefficients head-to-head rather than being scaled.

Data were scaled and standardized by removing the mean from each variable and scaling it to unit variance. To scale a variable $x$, we use

$$z = \frac{(x - \mu)}{s} \tag{1}$$

where $\mu$ is the mean of the samples and $s$ is the standard deviation. The scaler used for the training set should be the same scaler used for testing because the test set will necessarily have different means and standard deviations.

The flowchart in Figure 2 summarizes the data and methods used to estimate Sw in the MBM.

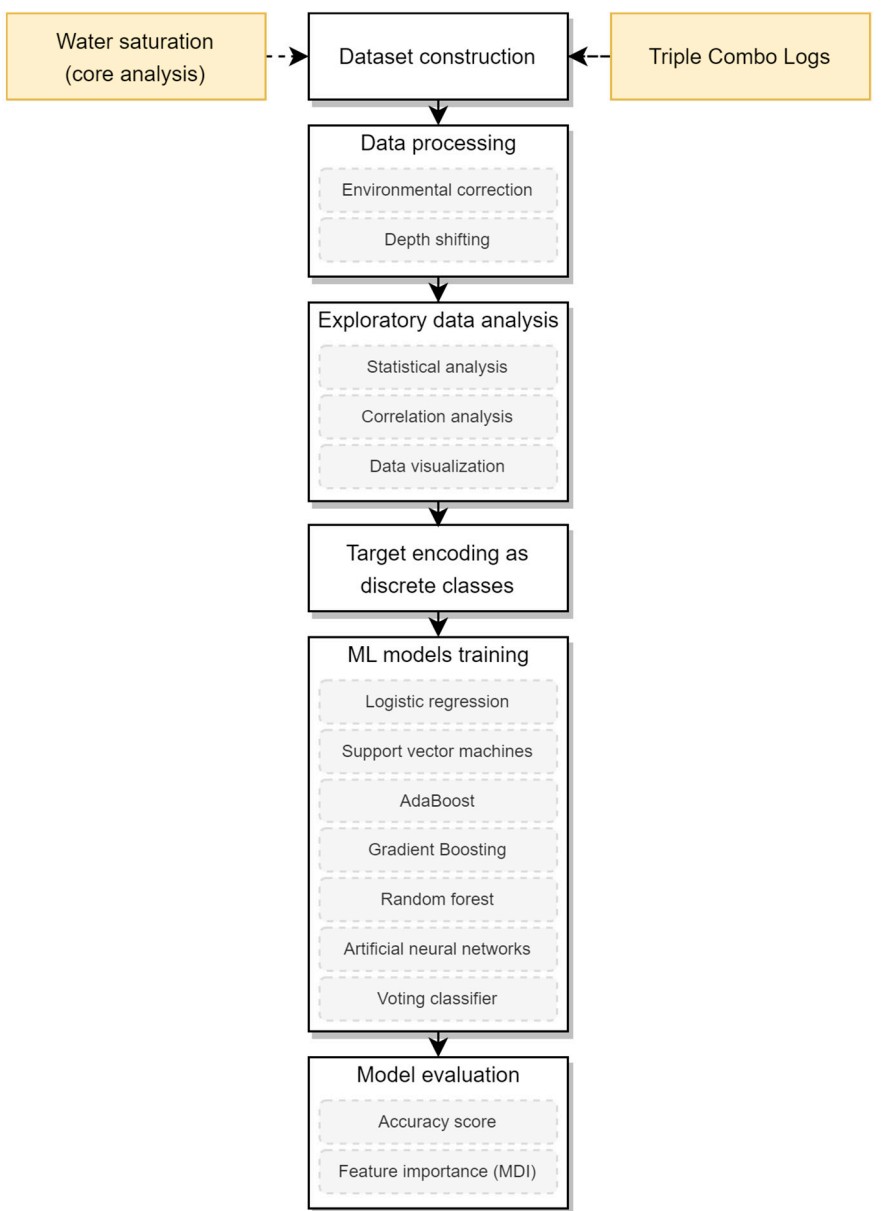

**Figure 2.** Flowchart for water saturation prediction from conventional well logs using ML models.

## 4. Results and Discussion

In this section, we present and discuss the results of seven ML models used to predict Sw in the MBM. ML models were applied due to the limitations of resistivity-based methods in such complex and heterogeneous reservoirs. Classification ML models were used over regression models due their low performance and the inherent uncertainty of Sw measured in the tight core samples of the MBM. The measurements could be either over- or underestimated.

The selected ML models were processed, trained, and tested with the objectives to:

- Investigate the potential linear relationship between well logs, Sw calculated using the modified Simandoux method, and Dean–Stark Sw.
- Assess the performance of the Sw prediction models and compare their accuracy with that of conventional methods.
- Determine the well logs that have the highest feature importance among the applied ML models.

In the following subsections, we conducted a petrophysical analysis, followed by an analysis of the data used for prediction, to finally apply and test the performance of the ML models. The results were compared with regression ML models and resistivity-based methods.

### 4.1. Petrophysical Analysis

Before training the selected ML models, we conducted a petrophysical analysis to calculate Sw using the Archie, Simandoux, modified Simandoux, Indonesian, Waxman–Smits, and dual-water methods (Figure 3). We evaluated the accuracy of the models using Sw measured with the Dean–Stark method. Before estimating Sw using resistivity-based methods, we calculated the clay volume (Vsh) and calibrated the results with the Vsh measured from the X-ray diffraction analysis. Effective porosity was also calculated and calibrated with porosity measured from cores, and Archie parameters (a, n, and m) were defined. From the petrophysical analysis and among all the models used to calculate Sw, the modified Simandoux method provided the most accurate estimation. The results of the model were then used as input to train and test the ML models. Figure 3 shows the results of Sw calculated using the resistivity-based methods and calibrated with core data.

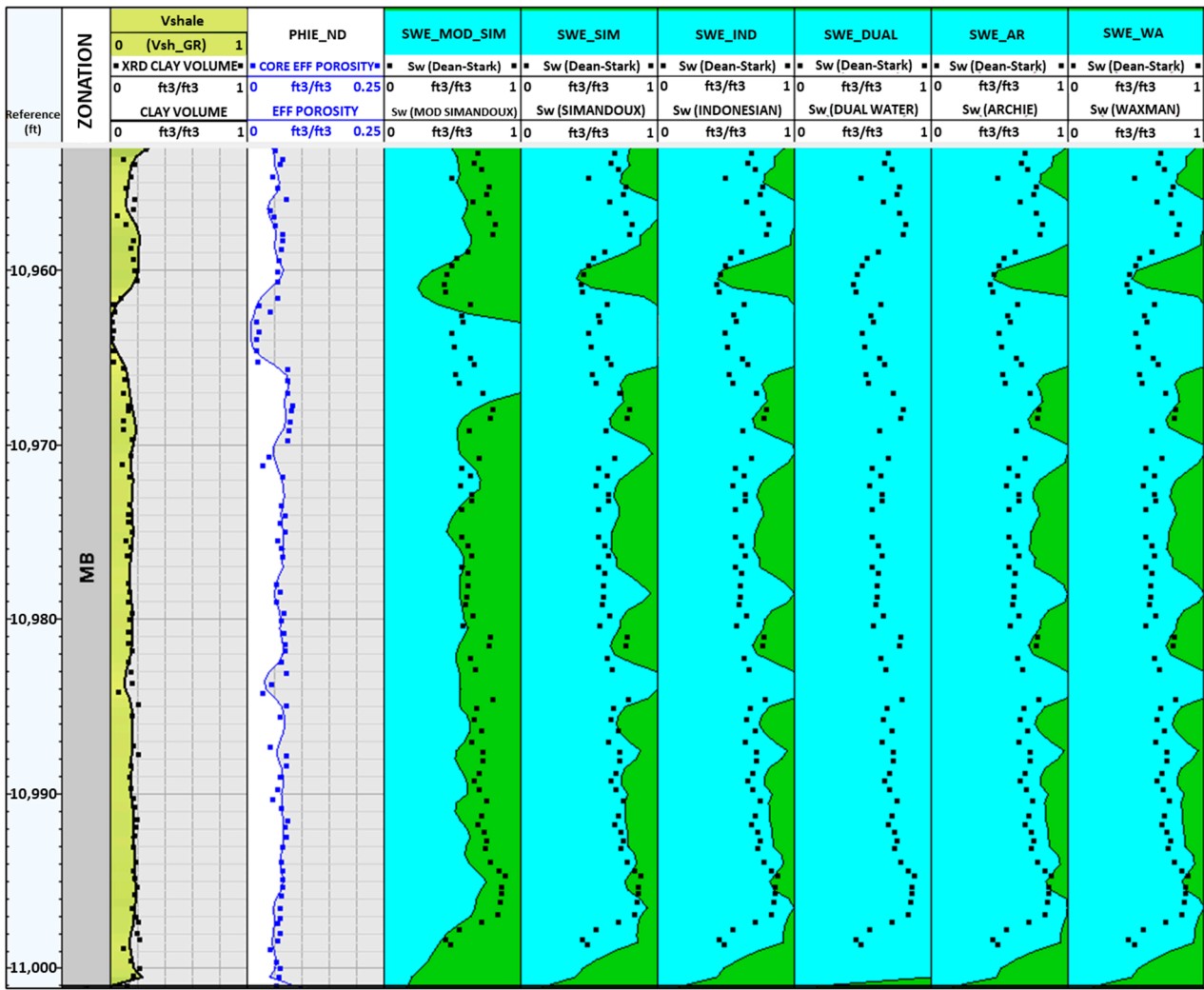

**Figure 3.** Petrophysical analysis of the Middle Bakken Formation. Tracks from left to right: track 1: reference depth, track 2: Thickness of the MBM, track 3: clay volume calibrated with XRD measured clay volume, track 4: effective porosity calibrated with core measured porosity, tracks 5 to 10: Sw calibrated with core measured Sw, oil saturation (green color), water saturation (blue color).

*4.2. Exploratory Data Analysis*

To investigate the relative importance of the input variables with the output variable, exploratory data analysis of the constructed dataset was conducted. This analysis enabled us to understand the dataset's patterns, linearity, and potential relationships. By examining measures of mean, standard deviation, and the Pearson correlation coefficient, we were able to identify outliers, discern trends, and establish a solid foundation for subsequent modeling and analysis techniques. Table 1 shows the different statistics of the predictive variables (i.e., well logs) and predicted variables (i.e., water saturation).

**Table 1.** Statistical summary of the predictive and predicted variables.

|  | **GR** | **R** | **RHO** | **NPOR** | **PE** | **Vsh** | **Por** | **Sw$_{Simandoux}$** | **Sw$_{core}$** |
|---|---|---|---|---|---|---|---|---|---|
| mean | 80.79 | 11.02 | 2.62 | 0.0865 | 3.59 | 0.1101 | 0.0818 | 0.3016 | 0.3836 |
| std | 16.56 | 18.84 | 0.03 | 0.0263 | 0.39 | 0.0393 | 0.0221 | 0.1248 | 0.1466 |
| min | 26.86 | 2.11 | 2.51 | 0.0108 | 2.61 | 0.0133 | 0.0278 | 0.0200 | 0.0540 |
| 25% | 73.25 | 4.59 | 2.60 | 0.750 | 3.33 | 0.0890 | 0.0675 | 0.2022 | 0.2593 |
| 50% | 80.89 | 6.44 | 2.62 | 0.0851 | 3.55 | 0.1063 | 0.0800 | 0.3151 | 0.3695 |
| 75% | 90.88 | 11.0 | 2.64 | 0.0993 | 3.83 | 0.1315 | 0.0958 | 0.3883 | 0.4690 |
| max | 124.47 | 254.43 | 2.69 | 0.1550 | 5.66 | 0.2435 | 0.1518 | 0.6682 | 0.7490 |

Figure 4 shows the distributions of the various variables and their distributions' kernel density estimates. Most of the variables show a normal distribution, with some exceptions. The photoelectric factor, modified Simandaux saturation, and core saturation exhibit a bimodal distribution, while resistivity demonstrates significant positive skewness with extreme values. These extreme values, located near the upper and lower Bakken members within the Middle Bakken, should not be considered outliers.

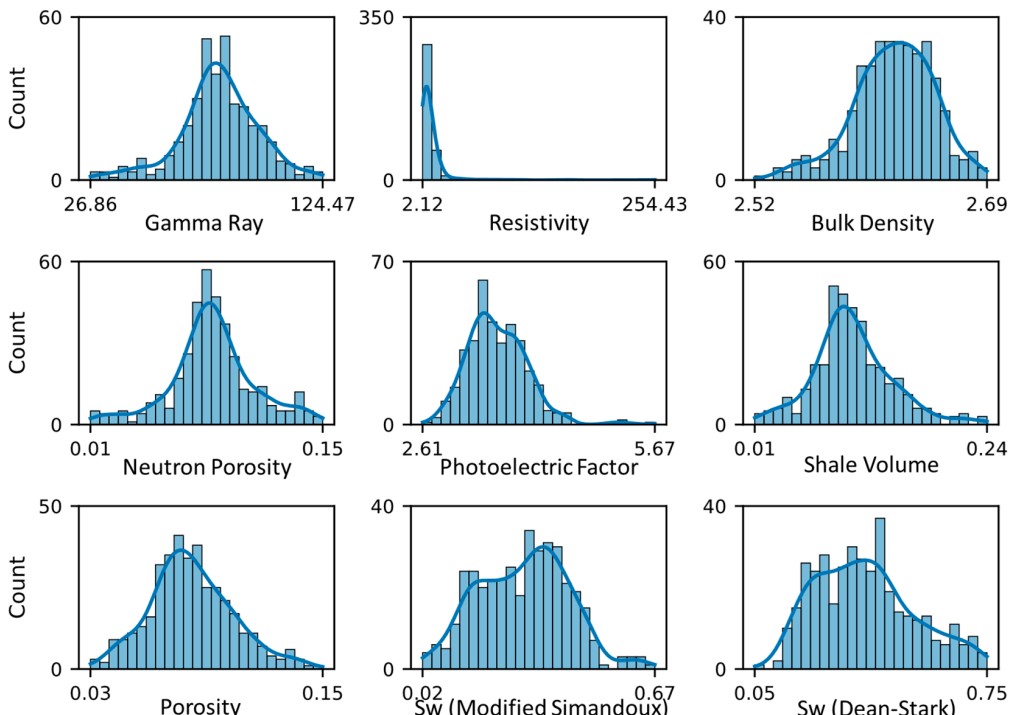

**Figure 4.** Histograms and skewed normal distribution curves of the input and output variables used to predict Sw.

Figure 5 displays the correlation matrix, providing insights into the relationships between the predictive variables (well logs) and the target variable. Notably, none of the predictive variables exhibit a strong linear correlation with the target variable. However, the Sw calculated from the modified Simandoux emerges as the variable with the highest linear correlation coefficient of 0.44, followed by bulk density at 0.36. While a high correlation

coefficient indicates a potentially robust and accurate ML model, the lack thereof only implies the absence of a direct linear relationship between the predictive and predicted variables. Nonetheless, there still can be nonlinear relationships that can be found and exploited by ML models.

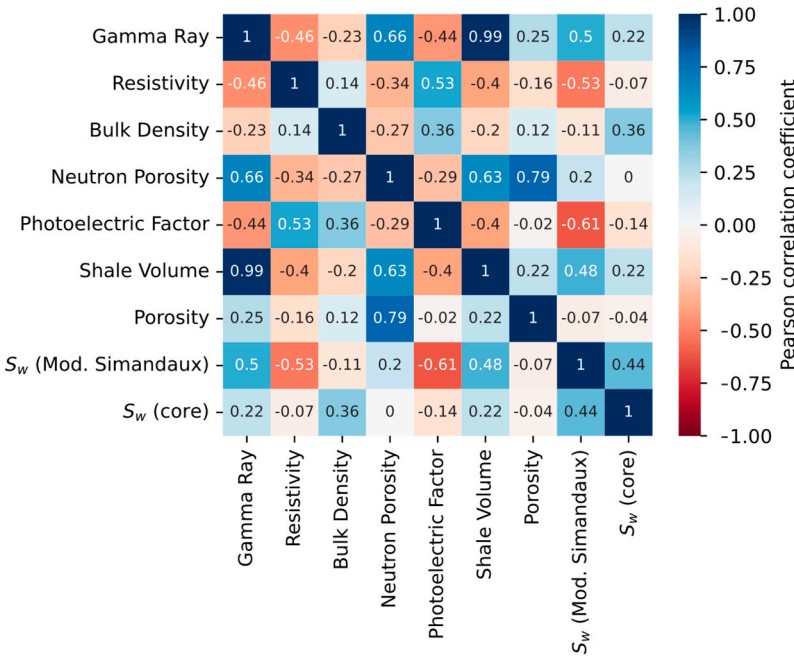

**Figure 5.** Correlation matrix of the input and output variables used for predicting Sw.

### 4.3. Machine Learning Model Performance

Figure 6 shows the accuracy scores of the various models on the test set. Among them, gradient boosting and random forest achieved the highest classification accuracy, 82.89% and 77.62%, respectively. The model parameters are summarized in Table 2. Using a soft voting scheme, these two models were subsequently combined into a voting classifier. This approach sums the predicted probabilities for each class and selects the class with the highest probability.

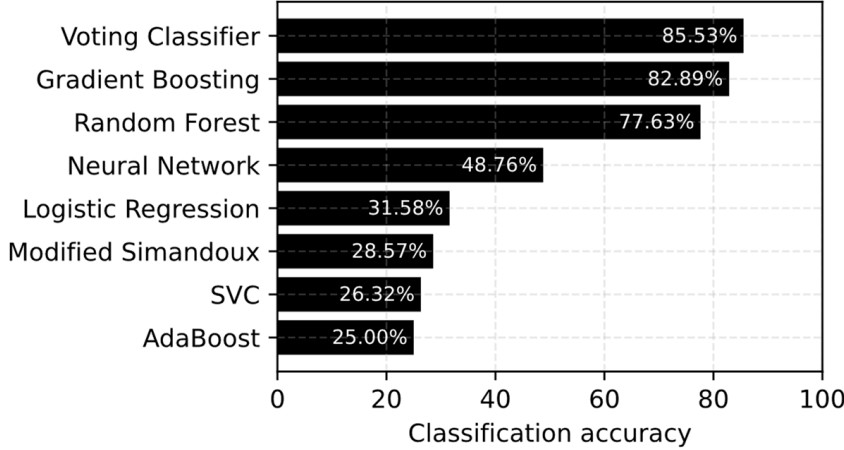

**Figure 6.** Classification accuracy of the applied ML models for the Sw prediction.

**Table 2.** Modeling parameters of gradient boosting and random forest classifiers.

|  | Loss | Learning Rate | Num. of Estimators | Criterion | Max Depth | Minimum Sample Split |
|---|---|---|---|---|---|---|
| Gradient boosting classifier | log_loss | 0.1 | 100 | friedman_mse | 3 | 2 |
| Random forest classifier | N/A | N/A | 100 | Gini | None | 2 |

The voting classifier yielded the best accuracy among all models, reaching a score of 85.53%. However, the ANN achieved a modest accuracy of 48.76% despite attempting various network architectures and hyperparameter tuning. This lower performance can be attributed to the relatively small dataset, which consisted of only 378 samples. The limited data may have hindered the network's training process and its ability to generalize to unseen data. Data collection was limited by the publicly available core analysis reports; if a large enough dataset is to be collected in future work, the artificial neural network should be able to have a better generalization, consequently achieving higher accuracy. Additionally, the modified Simandoux Sw, calculated for the entire dataset rather than just the testing subset, exhibited a low accuracy of only 28.57%.

*4.4. Model Evaluation*

The confusion matrices shown in Figure 7 present the results obtained from two methods for predicting Sw: the voting classifier and the modified Simandoux equation. The voting classifier exhibits high prediction accuracy, proving its efficiency in estimating Sw in the selected wells drilled in the MBM. The equation mostly overestimates the Sw, yielding significantly higher values than the Dean–Stark Sw.

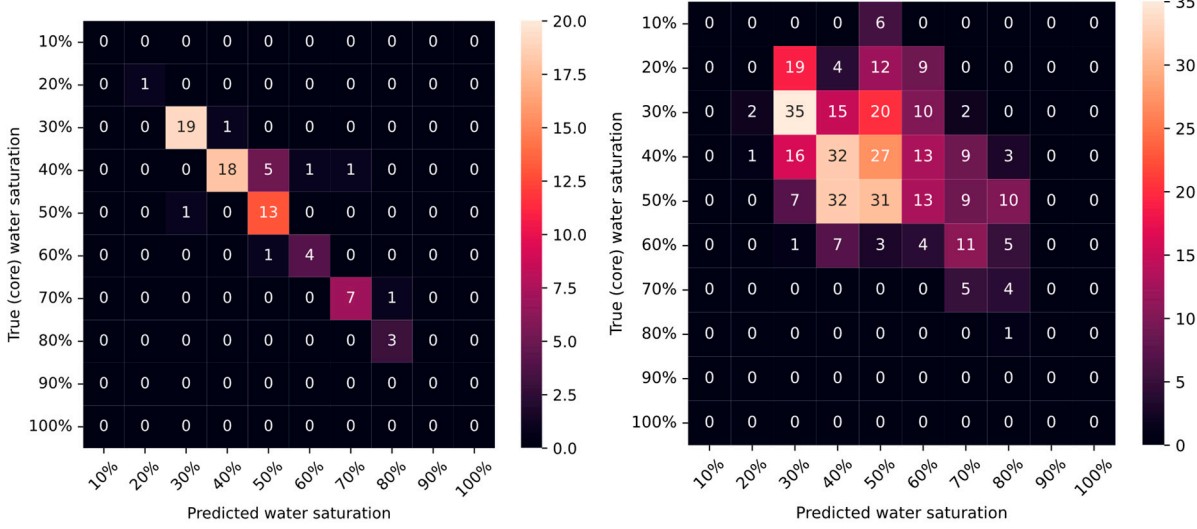

**Figure 7.** Confusion matrices of Sw predicted using voting classifier (**left**) and Sw calculated using modified Simandoux (**right**).

Despite its inaccurate estimation, modified Simandoux equation Sw calculations had the highest feature importance (measured by the mean decrease in impurity) for the ML model, followed by the resistivity log (Figure 8). The remaining logs had very similar feature importance scores. This indicates that the modified Simandoux calculations set the starting point for the ML models, which then used the well logs to refine the estimations and provide much more accurate predictions.

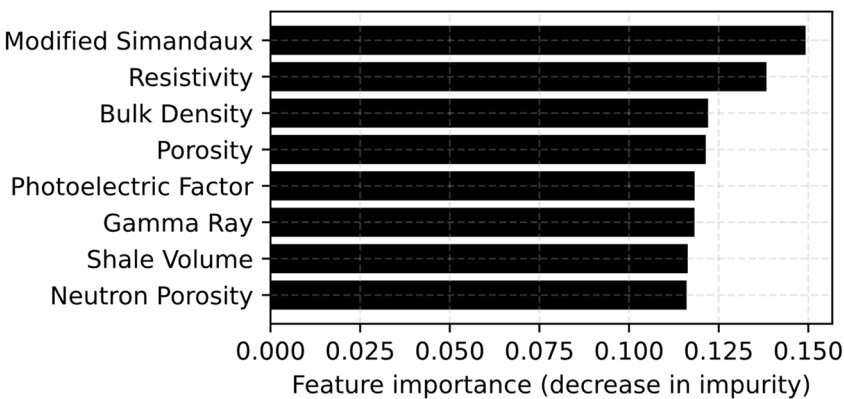

**Figure 8.** Feature importance of the different parameters used for Sw prediction.

Table 3 lists Sw prediction scores of various ML models from recently published studies using conventional well logs as input variables. The input variables used by these studies are identified with the R2 values achieved between predicted and measured Sw values. Note that all ML methods display a narrow range of high R2 values (ranging from 0.78 to 0.99; Table 2), but the SVM method achieved a lower R2 value (R2 = 0.78).

**Table 3.** Results from published studies that have used conventional well logs to predict Sw from ML models. The comparison was made with R2 values achieved by the models with the test subsets.

| Author | Samples and Wells Number | ML Model | Formation | Results |
|---|---|---|---|---|
| Ibrahim et al. [36] | 782 samples, 2 wells | ANN, ANFIS | Tight gas sandstone | $R^2 = 0.93$ |
| Hadavimoghaddam et al. [12] | 11 wells | XGBoost | Sandstone | $R^2 = 0.999$ |
| Miah et al. [10] | 182 samples | ANN and SVM | N/A (Bengal Basin) | $R^2 = 0.999$ |
| Khan et al. [37] | 150 samples | ANN and ANFIS | N/A (South Asian field) | $R^2 = 0.94$ |
| Hamada et al. [38] | 269 samples | ANN | Shaly sandstone | MSE = 0.012 |
| Gholanlo et al. [39] | 564 samples, 1 well | ANN | Carbonate | $R^2 = 0.87$ |
| Boualam et al. [9] | 2509 samples | SVM and ANN | Tight carbonate | $R^2 = 0.78$ |
| This study | 378 samples, 29 wells | Voting classifier | Ultra tight and multimineral formation | Accuracy = 85.53% |

In this study, the classification approach outperformed both the initially proposed regression approach and resistivity-based methods, resulting in higher accuracy. The limitations faced by the initially proposed methods in accurately predicting Sw can be attributed to the following challenges:

- High heterogeneity of the MBM, including the presence of extremely thin laminations and significant variation in the volume of cement minerals.
- The low resolution of logging tools cannot accurately represent the high variation of physical properties (bulk density, neutron porosity, photoelectric factor, and resistivity) of such formations, which are used as inputs for Sw prediction.
- The uncertainty associated with laboratory measurement of Sw in tight cores using the Dean–Stark method, which undermines the accuracy of Sw prediction using ML regression algorithms, even when models show a high correlation coefficient.

Considering these challenges, the application of ML classification models has proved to provide the most accurate estimation of Sw. The proposed workflow introduces a methodology aimed at improving the estimation of Sw in unconventional reservoirs. Notably, this approach takes into consideration the inherent heterogeneity of the formation and the limitations posed by logging measurements and the Dean–Stark method.

The application of classification models yields prediction results as classes representing ranges of Sw, rather than continuous values. The robustness of the applied models relies on using data from wells distributed across the MBM to capture the reservoir heterogeneity. To the best of our knowledge, this is the first attempt to predict Sw using classification ML models in reservoirs where regression ML models and resistivity-based methods fail to accurately estimate Sw.

To improve this workflow, it is essential to quantify the effect of the uncertainty associated with logging and laboratory core measurements on the predicted Sw. This will lead to more accurate classes for a representative Sw classification. Overcoming these limitations will yield a more precise and dependable Sw prediction in heterogeneous reservoirs.

In conclusion, this study offers an effective solution for estimating Sw in reservoirs where resistivity-based methods and ML regression models are underperforming. It also establishes a basis for future research, enabling validation and improvement of the proposed workflow. Additionally, the suggested workflow can be extended to predict other rock properties, such as absolute permeability in the Bakken, by integrating it with the approach developed by Aimen et al., 2022 [19]. Future research should replicate this workflow in diverse fields, including the Bakken Formation and other complex reservoirs. Moreover, exploring alternative classification ML models could further refine the prediction process and enhance the overall outcomes.

## 5. Conclusions

In this study, we assessed the performance of seven classification ML models in predicting the Sw of the MBM as a tight, low-resistivity, and multimineral reservoir. Well logs, Sw calculated using the modified Simandoux method, and Dean–Stark Sw data were used to train and test the models. The performance results were then compared with the accuracy of the regression ML models and resistivity-based methods. The results of the ML models led to the following conclusions:

- The voting classifier model, based on gradient boosting and random forest, displays the highest accuracy of Sw in the MBM.
- The Sw calculated using the modified Simandoux method tends to be overestimated in the MBM. However, using it as input to train and test the classification ML models improved result accuracy.
- Petrophysical data processing, which consists of depth shifting, environmental correction, and log normalization, is crucial for accurate prediction of Sw.
- The voting classifier model, based on gradient boosting and random forest, can accurately match the Dean–Stark Sw within a specific range. Therefore, it can be a viable alternative to expensive laboratory tests.
- We propose applying classification ML models to predict other rock properties, such as permeability and shale volume. This suggestion stems from the recognition that these properties share similar limitations as water saturation.

**Author Contributions:** Conceptualization, A.L. and I.M.; methodology, I.M.; software, I.M.; validation, V.R. and H.O.; investigation, I.M. and A.L.; writing—original draft preparation, I.M.; writing—review and editing, A.D. and O.B.; supervision, V.R. All authors have read and agreed to the published version of the manuscript.

**Funding:** This research was funded by the LeNorman Family Excellence Fund.

**Institutional Review Board Statement:** Not applicable.

**Informed Consent Statement:** Not applicable.

**Data Availability Statement:** Data available upon request.

**Acknowledgments:** The first author would like to acknowledge the scholarship provided by the LeNorman Family Excellence Fund.

**Conflicts of Interest:** The authors declare no conflict of interest.

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
