# Peer review of "Water Saturation Prediction in the Middle Bakken Formation Using Machine Learning"

_2673-4117, doi:10.3390/eng4030110_

Round 1
Reviewer 1 Report
The authors have presented a workflow to predict water saturation in ultra-tight, multimineral, low-resistivity reservoirs such as Middle Bakken Formation using well logs, core data, and 7 different machine learning classification algorithms. While they managed to develop a new approach to predict water saturation with relatively high accuracy, explained the details of the procedures thoroughly, and discussed the results adequately, there are some areas for improvement before this manuscript is ready to be published. There are some facts mentioned in the literature review, for which references should be provided. The grammar needs improvement throughout the entire manuscript. General details of machine learning algorithms are not the focus of this study and can be avoided. Additional required clarification, adjustments, and corrections are mentioned in the reviewed manuscript to help the authors. Most importantly, however, the final argument about the advantages of this study over previous similar studies needs improvement. The authors need to extend their discussion about why and how this study is adding values to the scientific communities. They need to be clearer and more specific about why one should select this approach over other similar approaches. This has been attempted to some level in the Introduction and Discussion, but stronger argument is needed to make this study bold enough to be published. More details to help the authors are provided in the reviewed manuscript.
N/A
Author Response
Response to Reviewer 1 Comments
Thank you for your valuable comments. Please find below the response to your comments.
Comment: The authors have presented a workflow to predict water saturation in ultra-tight, multimineral, low-resistivity reservoirs such as Middle Bakken Formation using well logs, core data, and 7 different machine learning classification algorithms. While they managed to develop a new approach to predict water saturation with relatively high accuracy, explained the details of the procedures thoroughly, and discussed the results adequately, there are some areas for improvement before this manuscript is ready to be published. There are some facts mentioned in the literature review, for which references should be provided. The grammar needs improvement throughout the entire manuscript. General details of machine learning algorithms are not the focus of this study and can be avoided. Additional required clarification, adjustments, and corrections are mentioned in the reviewed manuscript to help the authors. Most importantly, however, the final argument about the advantages of this study over previous similar studies needs improvement. The authors need to extend their discussion about why and how this study is adding values to the scientific communities. They need to be clearer and more specific about why one should select this approach over other similar approaches. This has been attempted to some level in the Introduction and Discussion, but stronger argument is needed to make this study bold enough to be published. More details to help the authors are provided in the reviewed manuscript.
Response: The comments were all adressed in the updated manuscript.
- In the introduction section, three papers were used as references to find research gaps and identify the problem statement where references to these papers have been added.
- We have also extended and improved the introduction and discussion sections, by adding more arguments on the advantages of using classification ML algorithms over regression ML algorithms and presenting the importance of the results for accurate reservoir characterization.
- Finally, we have improved the grammar throughout the entire manuscript.
Reviewer 2 Report
(Line – 19): “… inaccuracy and lack of robustness of the tried and tested regression models”. The author is talking about which models and related to whom? Kindly revise the sentence in such a way that the reader can understand it easily.
(Line – 26) “….. the prediction of Sw and other reservoir properties ….”. What other properties? In the manuscript only Water saturation Sw is discussed.
(Lines – 35 – 45) More references must be provided.
(Lines – 129 – 132) The depth taking as a reference may mislead the results because, in a field of reservoirs, the formations may dip on some angle so the formation at some certain depth may not exist (or another formation is there) at the same depth in the other well. Kindly review this and justify with logical reasoning to have the depth as a reference.
(Lines – 132 – 141) Preprocessing is mentioned but the author did not explain what methodology was adopted and how applied for the environmental correction, depth shifting, and log normalization.
(Lines – 14 – 148) Kindly explain “…. the models were still giving poor results….” How the author is assessing the result is poor. Only the “correlation coefficient” value shows that the results are poor, which is mainly showing the relationship agreement between the variables. Kindly explain.
(Line – 164) “…… are the different features …” What features? Related to this study then should be explained.
(Line – 197 – 208) More references must be added.
(Line – 219) Kindly mention what the 8 inputs and 32 neurons are used for clarity.
(Line – 225 – 232) In section 3.2.7, how the Voting classifier is related to this study? Kindly mention it for clarity.
(Line – 253) Here is a major deficiency in the manuscript that has to be managed. The author has not provided the mathematical models for 07 ML Models and has not provided the initial data (maybe an extract of 378 samples) to verify the claims. Without providing the model details (data also) it seems an incomplete manuscript.
(Line – 279) Figure 6 is presented but there is a connection mentioned in the manuscript text.
(Line – 279) Figure 6 has a column, showing the XRD data. In the manuscript, no data was presented or mentioned.
(Line – 279) What is the description of the samples (picture also) used for water saturation measurement in the Dean-Stark method? It is measured by the author or just data is acquired and used for analysis. The section shown in Figure 06 is 50 ft. So the physical samples are taken from this core section? What are the sizes of the plugs used for Sw measurement by the Dean-Stark method?
(Line – 162) “…thickness and the heavy heterogeneity…:” What you mean by heavy here?
(Line – 172) Figure – 4 is not readable. What is the reference? Kindly provide the detail and redraw it to make it readable.
The English quality of the overall manuscript is good. Some minor editing may be required.
Author Response
Response to Reviewer 2 Comments
Thank you for your valuable comments. Please find below the response to your comments.
Abstract
Comment 1: (Line – 19): “… inaccuracy and lack of robustness of the tried and tested regression models”. The author is talking about which models and related to whom? Kindly revise the sentence in such a way that the reader can understand it easily.
Response 1: We edited the sentence I line 19 for clarity
Comment 2: (Line – 26): “….. the prediction of Sw and other reservoir properties ….”. What other properties? In the manuscript only Water saturation Sw is discussed.
Response 2: We meant that although this prediction workflow was created to estimate water saturation, it can also be used to estimate other petrophysical properties such as porosity, shale volume, and water saturation in reservoirs where conventional methods have poor performance. However, we have edited the sentence and removed it to “and other reservoir properties” on page one line 26.
Comment 3: (Lines – 35 – 45): More references must be provided.
Response 3: We have added more references as requested
Comment 4: (Lines – 129 – 132): The depth taking as a reference may mislead the results because, in a field of reservoirs, the formations may dip on some angle so the formation at some certain depth may not exist (or another formation is there) at the same depth in the other well. Kindly review this and justify with logical reasoning to have the depth as a reference.
Response 4: The Middle Bakken reservoir varies in depth and thickness. However, the meaning of the sentence from lines 128 to 132 was that after defining the zonation of our studied reservoir in each of the selected wells, well logs and core data were selected. Due to some well-logging operational problems, the reading of the depth of well logs does not represent the true depth of the formation. Therefore, we take the depth of core slubs used to measure rock properties in the laboratory as a reference. Hence, well logs have to be shifted to align with the core depth. This alignment allows for direct comparison and integration of information between the two datasets.
Comment 5: (Lines – 132 – 141) Preprocessing is mentioned but the author did not explain what methodology was adopted and how applied for the environmental correction, depth shifting, and log normalization.
Response 5: We have elaborated on the preprocessing workflow for well logs based on your comment
Comment 6: (Lines – 14 – 148) Kindly explain “…. the models were still giving poor results….” How the author is assessing the result is poor. Only the “correlation coefficient” value shows that the results are poor, which is mainly showing the relationship agreement between the variables. Kindly explain.
Response 6: We were referring to the R2 score of the regression model which was indicating poor predictions by the models (it is now clearly mentioned). We are also aware that the R2 score is not the ultimate approach for model selection since it does not take the model’s degrees of freedom into account, nevertheless, it is extensively used in the literature for water saturation prediction and it allows for comparison against the other studies.
Comment 7: (Line – 164) “…… are the different features …” What features? Related to this study then should be explained.
Response 7: It was supposed to be a general discussion of the logistic regression model, and by features we meant the independent variables. It will be edited and mentioned explicitly for clarity.
Comment 8: (Line – 197 – 208) More references must be added.
Response 8: We have added more references
Comment 9: (Line – 219) Kindly mention what the 8 inputs and 32 neurons are used for clarity.
Response 9: “8 inputs” are the different features (i.e., well logs and calculated features), it is now mentioned. But the number (32) of neurons in the hidden layer is just our choice for the neural architecture. There are no hard rules to choose the neurons in the hidden layer, 32 seemed as a good choice because it is not too small for the model to underfit and not too large for the model to overfit considering the relatively small dataset.
Comment 10: (Line – 225 – 232) In section 3.2.7, how the Voting classifier is related to this study? Kindly mention it for clarity.
Response 10: Voting classifier is an ensemble model that takes predictions from other models and aggregates them following hard and soft voting schemes (discussed in the manuscript). It is not particular to this study, it is just an approach to combine predictions from several models and achieve even better predictions.
Comment 11: (Line – 253) Here is a major deficiency in the manuscript that has to be managed. The author has not provided the mathematical models for 07 ML Models and has not provided the initial data (maybe an extract of 378 samples) to verify the claims. Without providing the model details (data also) it seems an incomplete manuscript.
Response 11: The data used for predicting water saturation was sent to you.
Comment 12: (Line – 279) Figure 6 is presented but there is a connection mentioned in the manuscript text.
Response 12: This has also been edited in the manuscript
Comment 13: (Line – 279) Figure 6 has a column, showing the XRD data. In the manuscript, no data was presented or mentioned.
Response 13: The aim of the petrophysical analysis in Figure 6 is to show that water saturation cannot be accurately predicted using conventional petrophysical models To calculate water saturation using the models shown in Figure 6, shale volume and porosity must therefore be calculated and calibrated with porosity measured from core data and shale volume measured from XRD analysis.
Since shale volume (Vsh) was just an input used to calculate water saturation, and XRD data was used for calibrating Vsh, We thought that there is no need to mention this detail. However, if you believe that it is important to mention it, we will be happy to add details about the XRD data.
Comment 14: (Line – 279) What is the description of the samples (picture also) used for water saturation measurement in the Dean-Stark method? It is measured by the author or just data is acquired and used for analysis. The section shown in Figure 06 is 50 ft. So the physical samples are taken from this core section? What are the sizes of the plugs used for Sw measurement by the Dean-Stark method?
Response 14: There are various methods used for measuring the water saturation (Sw) from cores, one of which is the Dean-Stark method. We did not measure water saturation from cores. Instead, we obtained the results from 378.
We did not perform a laboratory experiment to estimate water saturation from cores. We do not therefore know the size of the cores used to measure water saturation.
Comment 15: (Line – 162) “…thickness and the heavy heterogeneity…:” What you mean by heavy here?
Response 15: Thank you for your comment, this should be line 106 instead of line 162. it was a mistake from the authors. We meant high heterogeneity. Heavy was replaced by high in the manuscript.
Comment 16: (Line – 172) Figure – 4 is not readable. What is the reference? Kindly provide the details and redraw it to make it readable.
Response 16: The picture is embedded as SVG, it is scalable and there is a reference

Round 2
Reviewer 1 Report
Almost none of the comments are addressed and suggested revisions have not been applied or explained.

Author Response
Response to Reviewer 1 Comments
Thank you for your valuable comments. Please find below the response to your comments.
Comment: The authors have presented a workflow to predict water saturation in ultra-tight, multimineral, low-resistivity reservoirs such as Middle Bakken Formation using well logs, core data, and 7 different machine learning classification algorithms. While they managed to develop a new approach to predict water saturation with relatively high accuracy, explained the details of the procedures thoroughly, and discussed the results adequately, there are some areas for improvement before this manuscript is ready to be published. There are some facts mentioned in the literature review, for which references should be provided. The grammar needs improvement throughout the entire manuscript. General details of machine learning algorithms are not the focus of this study and can be avoided. Additional required clarification, adjustments, and corrections are mentioned in the reviewed manuscript to help the authors. Most importantly, however, the final argument about the advantages of this study over previous similar studies needs improvement. The authors need to extend their discussion about why and how this study is adding values to the scientific communities. They need to be clearer and more specific about why one should select this approach over other similar approaches. This has been attempted to some level in the Introduction and Discussion, but stronger argument is needed to make this study bold enough to be published. More details to help the authors are provided in the reviewed manuscript.
Response:
Thank you very much fir the valuable comments to improve the manuscript. Below are the responses to your comments
Comment 1: There are some facts mentioned in the literature review, for which references should be provided.
Response 1: References were added in the introduction section (lines 34, 35, 41, 45, 48, and 51) to present:
- The challenges in producing from tight reservoirs (reference added in lines 34 and 35)
- The importance of accurately characterizing reservoir and fluid properties for better decision-making (reference added in line 39).
- Prove that predicting water saturation in heterogeneous reservoirs using resistivity-based methods and regression-based machine learning algorithms results in a poor estimate of the property (reference added in lines 41, 45, and 48).
- The accuracy of ML models is highly dependent on the quality of logs and the complexity of the studied reservoirs (reference added in line 51).
Comment 2: The grammar needs improvement throughout the entire manuscript.
Response 2: We improved the grammar throughout the entire manuscript, mainly in the introduction and results, and discussion sections. Please see the uploaded manuscript.
Comment 3: The final argument about the advantages of this study over previous similar studies needs improvement.
Response 3: We have improved our arguments (from line 373 to 390) by first mentioning the limitations of resistivity-based Sw methods and ML regression algorithms, and then listing the advantages of using ML classification algorithms for accurate water saturation estimation.
Comment 4: General details of machine learning algorithms are not the focus of this study and can be avoided.
Response 4: We briefly explained the machine learning algorithms applied and the difference between them before applying them for water saturation prediction.
The other reviewer asked us to keep this section of the paper, add more references and explain more how the voting classifier is related to this study.
Comment 5: The authors need to extend their discussion about why and how this study is adding values to the scientific communities.
Response 5: We have extended the discussion by first discussing the future development of the proposed workflow (from line 373 to 387) and improving the last paragraph to better explain the significance of the study to geoscientists and petroleum engineers working on the characterization of unconventional reservoirs (from line 396 to 403)
Comment 6: You need to be clearer and more specific about why one should select this approach over other similar approaches. This has been attempted to some level in the Introduction and Discussion, but stronger argument is needed to make this study bold enough to be published. More details to help the authors are provided in the reviewed manuscript.
Response 6: We improved the introduction and discussion and tried to be as clear as possible on the advantages of applying classification ML algorithms over regression ML and resistivity-based methods in unconventional reservoirs. (from lines 41 to 48, from line 81 to 103, and from line 373 to 387)
Comment 7: Additional required clarification, adjustments, and corrections are mentioned in the reviewed manuscript to help the authors.
Response 7: You mentioned that clarification, adjustments, and corrections are mentioned in the reviewed manuscript to help us improve it. However, when we download the review manuscript, we did not find these comments. If you have an updated one please resend it to us.

Reviewer 2 Report
I think author did a good job and manuscript is recommended for publishing.
Author Response
Thank you for your answer
Round 3
Reviewer 1 Report
The quality of the manuscript is significantly improved.
Author Response
Thank you very much for the valuable comments on improving the manuscript.